# Physical Activity: A Viable Way to Reduce the Risks of Mild Cognitive Impairment, Alzheimer’s Disease, and Vascular Dementia in Older Adults

**DOI:** 10.3390/brainsci7020022

**Published:** 2017-02-20

**Authors:** Patrick J. Gallaway, Hiroji Miyake, Maciej S. Buchowski, Mieko Shimada, Yutaka Yoshitake, Angela S. Kim, Nobuko Hongu

**Affiliations:** 1Department of Nutritional Sciences, The University of Arizona, Tucson, AZ 85721-0038, USA; gallaway@email.arizona.edu (P.J.G.); ask1@email.arizona.edu (A.S.K.); 2Nishinomiya Kyoritsu Neurosurgical Hospital, Hyogo 663-8211, Japan; miyake@nk-hospital.or.jp; 3Department of Medicine, Vanderbilt University, Nashville, TN 37232-5280, USA; maciej.buchowski@vanderbilt.edu; 4Chiba Prefectural University of Health Sciences, Chiba 261-0014, Japan; mieko.shimada@cpuhs.ac.jp; 5National Institute of Fitness & Sport in Kanoya, Kagoshima 891-2311, Japan; yositake@nifs-k.ac.jp

**Keywords:** aging, Alzheimer’s disease, cognitive decline, dementia, health, mild cognitive impairment, physical activity

## Abstract

A recent alarming rise of neurodegenerative diseases in the developed world is one of the major medical issues affecting older adults. In this review, we provide information about the associations of physical activity (PA) with major age-related neurodegenerative diseases and syndromes, including Alzheimer’s disease, vascular dementia, and mild cognitive impairment. We also provide evidence of PA’s role in reducing the risks of these diseases and helping to improve cognitive outcomes in older adults. Finally, we describe some potential mechanisms by which this protective effect occurs, providing guidelines for future research.

## 1. Introduction

The alarming rise of neurodegenerative diseases in the developed world is becoming one of the major medical issues affecting older Americans. By the year 2030, >20% of Americans will be over 65 years of age [1,2]. There are now an estimated 5.4 million Americans (one in nine), aged 65 years and older, with Alzheimer’s disease (AD), the most common form of dementia, and this number is expected to almost triple by 2050, as the population ages [3]. AD has become the sixth leading cause of death in the U.S., and the fifth leading cause of death among Americans aged 65 years and older, with deaths due to AD increasing by 66% between 2000 and 2008; in contrast to the decline of most other leading causes of death during the same period [4]. AD will likely become a more prominent health issue in developing countries in the near future, as life expectancies of those populations are longer, due to improved medical care. The World Health Organization (WHO) estimates that 47.5 million people are living with dementia and 7.7 million new diagnoses are made every year, worldwide [5]. Psychological disorders, such as depression, are also common, and are the second leading cause of disability in older populations [6]. The Italian Longitudinal Study on Aging (ILSA) provided evidence in a prospective, cohort study, that depression and physical disability in older adults have a complex relationship [7]. Perhaps the most alarming aspect of the increase in dementia cases, is that there are currently no cures or new effective therapies [8]. However, some lifestyle factors, such as physical activity (PA), could lower the risk of certain forms of dementia [9]. In a recent Lancet Series on PA, Progress and Challenges (2016), Sallis and colleagues reported that regular PA could prevent almost 300,000 cases of dementia per year, worldwide, if everyone were physically active [10]. In this article, our goal is to explore the role of PA in reducing the risks of age-related AD, vascular dementia (VaD), and mild cognitive impairment (MCI). We define dementia, AD, VaD, and MCI and describe clinical and research-based assessment tools used to diagnose each disorder. Finally, we provide evidence of PA’s protective effects of cognition in older adults, and discuss some of the potential mechanisms of the protective effects of PA, proposing suggestions to guide future research on PA intervention programs in order to reduce the burden of dementia, primarily through prevention and improved health care. This review was based on searches of the US National Library of Medicine (PubMed), Ovid MEDLINE, Google Scholar, and Web of Science, using terms to identify the risk exposure (physical inactivity or sedentary), combined with terms to determine the outcomes of interest (cognitive impairment or decline or disorders or AD or dementia or MCI or VaD). A search filter was developed to include only human studies.

## 2. Defining and Diagnosing MCI, Dementia, AD, and VaD

### 2.1. Mild Cognitive Impairment (MCI)

MCI is usually described as a transitional state between normal aging and dementia [11,12]. An individual with MCI experiences a cognitive decline that is not severe enough to significantly interfere with daily life, yet is worse than expected for one’s particular age [11]. Although studies on the MCI reversion to normality and MCI stability are limited, it is common for individuals with MCI to have no further deterioration of cognitive function for several years [13,14]. In a clinical setting, physicians may diagnose MCI based on self-reported symptoms or cognitive tests, but a wide variety of operational definitions of MCI have resulted in unreliability, although progress toward an objective standard continues to be made [15]. Cognitive tests, such as the mini-mental state examination (MMSE) [16] and the Montreal Cognitive Assessment (MoCA) [17], have been developed to help screen for MCI and/or dementia. Point scores in the MMSE range from zero (severe dementia) to 30 (lack of cognitive impairment) [16]. The MMSE provides a measurement of the overall cognitive functioning, including attention and orientation, memory, registration, recall, calculation, language, and the ability to draw, to health professionals and researchers, but does not give clinicians the ability to predict if those with MCI will experience further cognitive decline due to dementia [18,19]. The MoCA was specifically developed to assist with MCI diagnosis. In a validation study, it detected 90% of MCI cases with previously established criteria for MCI, a standardized mental status test, and subjective complaints about memory loss by participants or families, over at least six months. The MoCA significantly outperformed the MMSE, which detected only 18% of the MCI cases [17]. In addition, the MoCA has been shown to be a sensitive tool for cognitive impairment associated with other clinical conditions, such as Parkinson’s disease [20], Huntington’s disease [21], and multiple sclerosis [22]. Although progress has been made toward making the MCI diagnosis objective, the concept of MCI still remains somewhat vague and controversial, and issues such as clinical criteria for practitioners need to be investigated [23]. Since MCI is a syndrome with multiple etiologies, it is not considered a disease, but an aggregate of cognitive symptoms, attributable to either an underlying precursory stage of a serious disease, or an idiopathic acceleration of cognitive decline, when compared to a normal state [19]. To signify this distinction, MCI due to AD is typically separated as the prodromal stage of AD, and it is diagnosed using criteria specific to the early stages of AD, including specific biomarkers that are not evident in other causes of MCI [24]. Although MCI sometimes persists as only a minor annoyance if it is of the non-AD variety, more often than not, it progresses to AD or another form of dementia [25]. MCI has become an important risk factor and indicator of the early stages of dementia, that has resulted in the testing of medications meant for AD treatment [26]. With MCI serving as a link between dementia and normal aging [27], it can potentially help us investigate the effect of, not only medications, but also lifestyle programs, such as regular PA, social activity, and diet, on age-related cognitive decline and dementia—both for improving cognition and delaying cognitive decline [28,29,30].

### 2.2. Dementia

Dementia is not a specific disease, but rather it refers to the symptomatic outcome of a number of serious neurodegenerative diseases that adversely affect cognitive function. Worldwide prevalence of people with dementia is estimated at 47.5 million, and is expected to double by 2030 and triple by 2050 [5]. Most patients with dementia display behavioral and psychological symptoms [31], such as memory loss and difficulty organizing or planning, and psychological changes, such as personality changes (aggression—verbal/physical), agitation, anxiety, depression, social withdrawal, and hallucinations [32]. Dementia is the result of serious neurodegeneration in the brain, significantly hinders daily activities, and can require a complete reliance on caregivers in later stages [32]. The most common form of dementia is AD, which causes 50%–75% of all dementia cases [33]. VaD is the second most common primary cause of dementia—at least 20% of dementia cases are due to VaD, and it is often present alongside AD or another form of dementia [34]. Although a definitive percentage is not available, due to many cases of dementia going undiagnosed, AD and VaD are estimated to be responsible for 70%–95% of dementia cases. Dementia is typically diagnosed by a healthcare provider in a clinical setting, by determining the extent of cognitive impairment, although this can be a difficult task due to dementia’s progressive nature—there is a range of severity in symptoms, depending on how far the disease has progressed when the patient is examined, and this largely relies on the examining physician’s discretion [35]. Positron emission tomography (PET) and single photon emission computed tomography (SPECT) are both brain imaging methods that are most commonly used to make a dementia diagnosis by examining the physical condition of the brain [36]. However, even with these tools, the difficulties of recognizing and diagnosing dementia are apparent, and approximately half of dementia cases are currently undiagnosed [37]. The distinctions between regular aging and the first signs of dementia may be difficult for a clinician to distinguish, and PET and SPECT scans may be expensive, which could discourage widespread testing. The changes may also happen slowly and subtly, and may thus be indiscernible to family members or caretakers. Currently, there is no cure for dementia.

### 2.3. Alzheimer’s Disease

Alzheimer’s disease (AD) has dramatically risen in the last couple of decades, becoming the sixth leading cause of death in the U.S., and the fifth leading cause of death among older adults, with deaths due to AD increasing by an astounding 66% between 2000 and 2008, in sharp contrast to the general decline of most other leading causes of death during the same period of time [38]. Alzheimer’s disease is a neurodegenerative, dementia-causing disease, with no known cure. Around 70% of AD cases occur after the age of 65 [39]. Amyloid-beta (Aβ) is a polypeptide—a chain of amino acids that is a protein precursor—that can build up on brain cells, causing plaques that are found in abundance in AD patients; these Aβ plaques are thought to be one of the major contributors to dementia caused by AD [40]. In addition to plaques, AD also appears to be related to tangles in the brain, which are structural abnormalities due to defective or deficient tau proteins; tau proteins support microtubules, which help provide cell structure and movement. AD is typically diagnosed by biomarkers, such as Aβ in cerebrospinal fluid, tau proteins, and regional brain volumes; these measurable substances can be used to predict AD progression in patients with MCI. However, cognitive markers, such as the symptoms reported by an examining physician, appear to be more effective predictors of the future development of AD, especially at baseline MCI, when it is first noticed and diagnosed [41]. To recognize the asymptomatic pre-clinical stage (up to a decade before the clinical onset of AD), standardized criteria for preclinical [42] and prodromal (amnestic MCI) stages [43] of the diagnostic criteria for AD, have been recommended for both clinical and research purposes.

### 2.4. Vascular Dementia

Vascular dementia (VaD) is the second most common form of dementia after AD, and is the result of impaired blood supply to the brain, which damages brain tissue when oxygen and nutrients are cut off. There are a number of possible causes of VaD [44]. It is often the result of a number of small, focal cerebral infarcts (small strokes) that may go unnoticed individually, but have an additive detrimental effect as more and more small areas of the brain are destroyed by ischemic events; however, there are also a number of other causal subtypes of cerebrovascular disease [45]. VaD may also be present along with other forms of dementia, such as AD, which can further complicate the condition, aggravating dementia symptoms, as access to more areas of the brain are lost. Diagnosing VaD is therefore no simple matter; there is currently a lack of validated criteria for establishing a diagnosis, and many of the various pathologies that reduce the brain’s blood supply are complex [46]. Although cerebrovascular lesions can be seen using brain imaging techniques, the diagnosis of VaD remains difficult, since such lesions may or may not be contributing to dementia symptoms, and this can lead to over-diagnosis of VaD as the cause of dementia [47].

### 2.5. Progression of a Neurodegenerative Disorder

Age-related neurodegenerative disorders, such as AD and VaD, generally show the same patterns in their progressions from normal aging to dementia. First, symptoms of MCI develop. MCI symptoms may improve if the condition is transient, and the patient may even go back to experiencing normal aging. The subject may also experience and remain in the MCI stage, without the condition progressing to dementia [13,14,25,26,27]. However, if there is an underlying cause, such as AD or VaD, MCI will eventually progress to dementia, and this step is irreversible. Intervention before irreversible brain damage occurs, is the best clinical practice for reducing the impact of dementia [28,29,30].

Numerous studies have revealed the connections between frailty, a pathological aging process that is reversible [48], and neurodegenerative disorders [49]. Cognitive frailty was first proposed by Panza and colleagues, who reported the risks of decreased cognitive functions, modulated by vascular factors [50]. In 2013, the International Academy on Nutrition and Aging and the International Association of Gerontology and Geriatrics, defined cognitive frailty as the heterogeneous clinical syndrome condition in older adults with both physical frailty and cognitive impairment, but excluding those with AD and other dementias [51]. Cognitive frailty is further refined into two subtypes; reversible and potentially reversible cognitive frailty [52]. The cognitive impairment of reversible cognitive frailty is subjective cognitive decline (SCD), a type of cognitive decline that may appear as the first symptom of preclinical AD, and/or positive biomarkers resulting from physical factors. [52]. MCI is the cognitive impairment of potentially reversible cognitive frailty. Recently, a longitudinal population-based study reported that reversible cognitive frailty in older adults increased the risk of developing dementia, particularly VaD, but not AD, and all-cause mortality [53]. The authors suggested that older adults with reversible cognitive frailty could benefit from a cognitive impairment intervention that may include regular PA, diet (e.g., Mediterranean diet), smoking cessation, and an active social lifestyle [53]. More research is required to determine the clinical screening criterion of cognitive frailty and the effectiveness of interventions for individuals with other geriatric disorders.

## 3. Risk Factors for AD, VaD, and MCI

There are a number of risk factors for AD, the most obvious of which is advanced age. In addition to environmental factors, genetic causes are implicated, as several genes that have been associated with AD; the most undisputed and well-known of which is the gene that encodes apolipoprotein E (APOE). The APOE gene is the strongest genetic risk factor for the development of late-onset AD, which accounts for >95% of all AD cases [54,55,56]. Many epidemiological studies suggest that the APOE ε4 allele carrier status of individuals have associations between modifiable lifestyle risk factors, and dementia and AD [57,58,59,60]. Accumulating data suggest the APOE ε4 allele plays an important role in Aβ plaques and clearance, tau protein tangle formations, oxidation, neurotoxicity, and dysfunction in lipid transport, which are the major hypotheses of AD pathogenesis [61]. The epidemiological population-based study which had an average of 21 years of follow-up times before the diagnosis of dementia, showed that APOE ε4 allele carriers’ risk of dementia may be more affected by lifestyle factors such as PA, dietary fat and fish oil intake, alcohol drinking, and smoking. The authors suggested that adopting a healthy lifestyle, including increased PA, should be utilized as a major preventive strategy to decrease the risk or postpone the onset of dementia among APOE ε4 allele carriers [62]. The genetics of AD is advancing quickly, as new AD-related genes continue to be found [63]. However, to date, no therapeutic interventions targeting the APOE or other genes have been successfully established [64,65].

Cardiovascular disease (CVD) is another apparent risk factor for AD, particularly peripheral arterial disease—a form of atherosclerosis [66,67]. According to a number of studies, depression in later life also presents a significant risk for acquiring AD [68,69]. Within a three-year period, depressed MCI patients had a twofold higher risk of developing AD than non-depressed MCI patients [70]. Hypertension [71], diabetes [72], hypotension [73], and hypercholesterolemia [74], are all additional risk factors for AD [75,76]. Poor sleep quality and sleep deficiency may present yet another risk factor, as sleep deprivation induces more Aβ buildup in the brain, while adequate sleep reduces it—moreover, Aβ buildup may also cause poor sleep patterns and increased wakefulness [77]. Poor sleeping patterns may therefore lead to increased Aβ formation, to further deteriorate sleep quality, resulting in positive feedback that may increase AD risk. There are a number of risk factors for VaD that are also risk factors for AD, including hypertension, diabetes, excessive adiposity, and dyslipidemia [78]. Although the effect of these conditions on VaD may be obvious due to their cardiovascular or metabolic nature, the link to AD may be more indirect, as the resultant disruption of vascular functions in the brain could be compounding the neurodegeneration caused by the Aβ plaques or neurofibrillary tangles of AD, by disrupting cerebral autoregulation [75,76,79]. The risk factors for MCI are generally the same as those for AD—this is to be expected because MCI can also be the prodromal stage of AD. The most significant risk factors for MCI are older age and hypertension [80]. Now that we have addressed the potential risk factors for the most common age-related neurodegenerative diseases, we can explore the role of PA as a protective or risk-reducing behavior.

## 4. Physical Activity’s Effect on Future Risk of MCI, Dementia, AD, and VaD

PA is beneficial for both physical fitness (e.g., changes in the cardiovascular system, bone and muscle) and mental health (e.g., emotional functioning—depression, moods, cognitive functioning, social functioning) in almost all older adults [9,81], and its effects have been extensively studied in healthy older adults and those with cognitive impairment, including MCI, dementia, AD, and VaD.

### 4.1. Long-Term Cognitive Effects of PA on Healthy Older Adults

PA throughout one’s life can enhance cognitive function later in life, so it should be encouraged at every age. In contrast, sedentary behaviors, such as viewing television for extended periods over the course of years, can negatively affect cognitive function later in life. A longitudinal study over 25 years, that measured the PA and television viewing habits of 3247 healthy adults (aged 18–30 years at the start of the study), found that higher levels of PA and lower amounts of television viewing, resulted in significantly better processing speeds and executive functions in cognitive tests at midlife [82]. Moreover, those who were physically active in midlife had a reduced risk of developing depression in late life [83]. Depression in late life has also been linked to dementia, particularly in those carrying the ε4 variant of the APOE gene that predisposes to AD in depressed individuals [84]. According to the findings of a 10-year longitudinal study involving 470 participants aged 79–98 years, PA can also help engage older adults in cognitive and social activities, which may be one of the factors that helps prevent cognitive decline [85]. Ideally, all adults should remain physically active throughout life [10,86], starting at a young age, to achieve optimal cognitive health as an older adult. However, there may be shorter-term benefits to increased PA levels [87,88,89,90].

### 4.2. Short-Term Cognitive Effects of PA on Healthy Older Adults

For healthy older adults, the short-term effects of PA need further investigation. One study found that the cognitive functions of memory and independence were improved in older adults (aged > 75) by a single session of low-intensity, range-of-motion exercise, but the effect might also be short-lived [87]. Another study found that healthy older adults practicing Tai Chi, or simple stretching and toning exercises, can improve global cognitive function, improve recall, and reduce subjective cognitive complaints after a one-year intervention [88]. Certain exercises, such as range-of-motion and Tai Chi, may show short-term benefits in cognitive ability in healthy older adults [87,88]. Aerobic exercise has been shown to attenuate cognitive decline, reduce brain atrophy, and improve physical health in healthy older adults [89,90], although there is some evidence that short-term cognitive effects are not as pronounced [89]. A review of 12 trials lasting from eight to 26 weeks, and included 754 older adults with no cognitive impairment, showed no short-term cognitive benefits from aerobic exercise [89]. However, the age of the older adult may also factor into the efficacy of aerobic exercise. One study showed that adults between 60 and 70 years old displayed significant cognitive benefits in spatial object recall and recognition from a three-month aerobic PA intervention that increased hippocampal perfusion—blood flow to the hippocampus. However, the positive effect of aerobic PA on perfusion may decline with age [91].

In recent years, there has been growing interest in resistance training (e.g., weight lifting, strength training) to improve cognition [92,93,94,95] and prevent brain volume loss in older adults [96]. The long-term impact on cognition and white matter volume in older women was reported in a 52-week randomized clinical trial of resistance training program that included machine exercises Keiser pressurized air system, free weights, non-machine exercises, or balance-and-toning training program that included stretching, range-of-motion, core-strength, balance, and relaxation exercises. Both interventions were performed twice per week. The resistance training program promoted memory, reduced cortical white matter atrophy, and increased peak muscle power after 2-year follow-up, relative to the balance-and-toning training program [96]. Also, it has been reported that cortical white matter volume is reduced among older adults with dementia, when compared to their healthy counterparts [97]. Thus, maintaining cortical white matter volume might be important for maintaining cognitive functions in older adults [96].

There are several studies supporting the hypothesis that resistance training has similar positive effects on cognitive function among older adults, to aerobic-based exercise training. However, there is no clear consensus on the underlying mechanisms by which resistance training promotes cognitive function and brain tissue integrity [9,95,96]. More research is needed to examine the variables of resistance training (i.e., intensity, frequency) and the possible mechanisms by which resistance training may prevent cognitive decline. A recent meta-analysis found that combined aerobic exercise and resistance training, had greater effects on reducing cognitive decline than these programs alone [98]. Current understanding of how PA promotes cognitive function is largely from aerobic training studies. Further research is required to identify the underlying changes in the body and brain (e.g., changes in brain volume) that improve cognitive function, using neuroimaging, physiological assessment, and circulating levels of various neurotrophins (e.g., IGF-1). Subsequent research should also determine how and what types of PA could help both healthy and frail older adults gain cognitive benefits in social and environmental settings of daily life.

### 4.3. Effects of PA on the Risk of Developing MCI and Dementia in Older Adults

PA is effective in reducing risk for developing MCI in older adults, but the optimizing of exercise training (i.e., types of PA, intensity, duration), cardiorespiratory fitness, age, level of cognition, medications, and social environments, may all play roles in the outcome [9]. The studies mentioned above showed that stretching and toning exercises hold the most promise for improving cognitive function in the healthy adults aged 75 years and older [87,88], while aerobic [90,91,99] and resistance exercise [92,93,94,95] also have positive effects. It seems that the risk of developing MCI may also be improved by exercise intensity. In older adults (aged ≥ 65 years), moderate exercise was shown to reduce the risk for MCI, while vigorous or light exercise did not show similar effect [100]. Increased PA in older adults also appears to reduce the risk of dementia due to AD and VaD, although more research must be done to explore the mechanisms of this effect [90,95,96]. In one study, adults of 65 years and older, participating in the Cardiovascular Health Cognition Study who regularly participated in four or more physical activities per week, had about half the risk of developing dementia as those participating in zero to one physical activities. However, those carrying the ε4 variant of the APOE gene—the greatest genetic risk factor for late onset AD—the risk was not affected by PA levels [101]. The frequency of the APOE ε4 gene is 19.0% in African American, 13.6% in Caucasian, 11.0% in Hispanic, and 8.9% in Japanese populations [102]. As mentioned earlier, for the majority of the population—those without the APOE ε4 gene—PA may have protective effects against the development of dementia [62]. This finding is supported by other studies showing a significant reduction of dementia found in older adults who exercised three or more times per week, when compared to those who did not [103]. Nevertheless, the mechanisms by which different types and frequencies of PA reduce dementia risk in older adults warrant further research.

### 4.4. Effects of PA on Older Adults with Cognitive Impairment

For older adults who have already developed a form of cognitive impairment, whether mild, such as those with MCI, or moderate to severe, as with dementia, PA can improve cognitive function, when compared to those with cognitive impairment who are not physically active [104]. Studies show that six to 12 months of exercise for those with MCI or dementia results in better cognitive scores than sedentary controls [105]. The positive effect of PA on cognitive function may be more apparent in older adults with MCI than in those with dementia, according to one review, but this may be due to the methodological issues of the performed studies; thus, more research is needed on the effect of PA on cognitive function in older adults with dementia [105]. However, the meta-analysis showed that aerobic exercise helps improve cognition in older adults with both AD and non-AD dementia, when combined with other standard medical treatments for dementia, and higher frequency interventions did not result in additional effects on cognition [106]. The results offer supporting evidence that PA intervention, with or without pharmacotherapy, is beneficial for cognition in patients with dementia.

## 5. Potential Mechanisms for PA’s Protective Effects

While the protective effect of PA on the aging brain is supported by numerous studies, the exact mechanisms are less clear. A recent review examined many possible mechanisms for how PA is linked to a reduced risk of age-related cognitive impairments, including MCI, AD, and VaD [107]. In this section, we will examine some of these potential mechanisms.

### 5.1. Increasing Blood Flow to the Brain

PA can increase blood flow to the brain, both during and shortly after a PA event, in response to increased needs for oxygen and energetic substrate [108,109]. The increased brain/cerebral blood flow triggers various neurobiological reactions, which provide an increased supply of nutrients. Moreover, cerebral angiogenesis—the development of new blood vessels in the brain—is increased by PA, and the brain’s vascular system is plastic, even in old age [110]. The increased vascularization of the brain, as well as the regular increases in blood flow that periods of PA provide, may reduce the risks of MCI and AD, by nourishing more brain cells and helping to remove metabolic waste or AD-inducing Aβ. The potential risk-reducing effect of increased blood flow on the development of VaD is more obvious, since VaD involves an impaired blood supply to the brain, while PA increases the blood supply. For instance, if a small artery in the brain is occluded by an embolus, this can often lead to an ischemic stroke, but blood can sometimes reach affected brain cells from an alternative path. Increased vascularization in the brain from PA can increase these alternative sources of blood during arterials occlusions, possibly limiting the damage [111]. There is strong evidence that cerebrovascular health may also play a large role in the severity of AD, as cases with brain infarctions, in addition to AD, showed worse symptoms of dementia than those with only AD [112]. More research is needed to determine the extent to which increasing vascularization of the brain may help reduce the risk of age-related neurodegenerative diseases.

### 5.2. Improving Cardiovascular and Metabolic Health

Earlier, we discussed how hypertension is one of the main risk factors for MCI, AD, and VaD [71,79]. Hypertension can increase the risk of strokes, as well as small strokes that are often the cause of VaD. Since strokes can complicate AD and aggravate dementia symptoms, it follows that hypertensive individuals could benefit by lowering their blood pressure, regardless of their level of cognitive impairment. Even low-intensity PA for 30 min, three to six times a week for nine months, can significantly lower blood pressure in elderly adults [112]. Because hypertension is a prominent risk factor, lowering blood pressure may be one of the mechanisms by which PA reduces the risk of many age-related neurodegenerative diseases. Diabetes is also a very significant risk factor for MCI, AD, and VaD [72]. The excess blood glucose levels found in those with diabetes causes tissue damage [113], inflammation [114], and microvascular disease [115], which possibly affect brain tissue, consequently increasing the chance of stroke. Regular PA can prevent type 2 diabetes and also helps manage blood glucose levels in those with diabetes [116]. By reducing the risk of diabetes and improving health conditions, improvement of metabolic health may be a secondary mechanism by which PA decreases the chances of MCI, AD, and VaD. Another risk factor for cognitive impairment in older adults that we have previously discussed is hyperlipidemia, or abnormally high lipid (cholesterol and triglycerides) blood concentration [74,78]. Regular PA can increase blood level high-density lipoproteins (HDL) that help carry cholesterol out of the bloodstream and into the liver, reducing hyperlipidemia. Thus, it seems plausible that PA reduces neurodegeneration risk through general cardiovascular and metabolic health improvement [117].

### 5.3. Preventing and Treating Depression

Depression is a known risk factor for developing dementia. Depression also appears to reduce certain cognitive functions in adults who are otherwise not cognitively impaired [118]. Although midlife depression doubles the chances of acquiring dementia later in life, it is harder to distinguish whether late-life depression is a risk factor for dementia, or vice versa; it could be a result of the early stages of dementia or MCI [119]. PA is effective for both treating and preventing depression [120], and therefore, it would stand to reason that reducing depression could be one of the means by which PA reduces the risk of AD, MCI, and other cognitive disorders. Even in cases where late-life depression is caused by the early dementia symptoms, PA could still be encouraged as part of a treatment for the depressive symptoms. Future research is needed to determine how health care professionals can deliver a PA program to healthy or physically/cognitively frail older adults. There is a need to determine the external factors of PA intervention, such as timing, dose, type, structure, and use of mindfulness that best ameliorate depression, and thereby reduce the risk of cognitive decline and prevent dementia.

### 5.4. Improving Sleep Quality

PA is associated with a reduction of insomnia symptoms, and other sleep quality and quantity problems, including problems with sleep onset (being able to fall asleep quickly) and sleep maintenance (staying asleep throughout the night) in older adults [121]. We previously noted a study showing that poor sleep quality is a potential risk factor for AD in particular [61], as sleep disturbances occur frequently in older adults with dementia [122] A specific mechanism by which sleep may reduce the risk of AD is through metabolic waste clearance in the brain, that occurs during sleep; this process also clears Aβ from the brain [123]. Aβ buildup results in plaques that contribute to AD, so clearing it from the brain during the deep stages of sleep may be the reason why adequate sleep reduces the risk of developing AD. This may also be a potential mechanism of the protective effect of PA on brain health in older age; PA promotes better sleep quality, which may in turn, helps clear harmful wastes, such as Aβ from the brain, thereby reducing the risk of dementia.

## 6. Conclusions

Several studies demonstrate the protective effect of PA on brain health, particularly by reducing the risk for the neurodegenerative dementia-causing diseases, AD and VaD, as well as their precursor, MCI. We have described potential direct and indirect mechanisms of this protective effect. More research is warranted to explore the relationships between PA and the aging brain. Other factors, including genetics, may affect the development of neurological disorders. However, in most cases, moderate PA is beneficial for both physical and mental health in older adults. Moderate-intensity aerobic exercise, resistance training, stretching, toning, and a range of motion exercises, may yield cognitive benefits in older adults. Although the exact mechanisms by which PA decreases the risk for dementia is not fully understood, PA should be encouraged [124], since it improves the quality of life for all older adults. The existing evidence shows that rates of dementia could be reduced, if people were physically active [10]. There is a possibility that PA may become the most important behavioral factor in facilitating healthy mental and physical aging (Figure 1). Current evidence supports PA’s short and long term cognitive benefits, regardless of age. Many mechanisms responsible for the PA’s protective effect against age-related cognitive impairment are still not fully understood.

## Figures and Tables

**Figure 1 brainsci-07-00022-f001:**
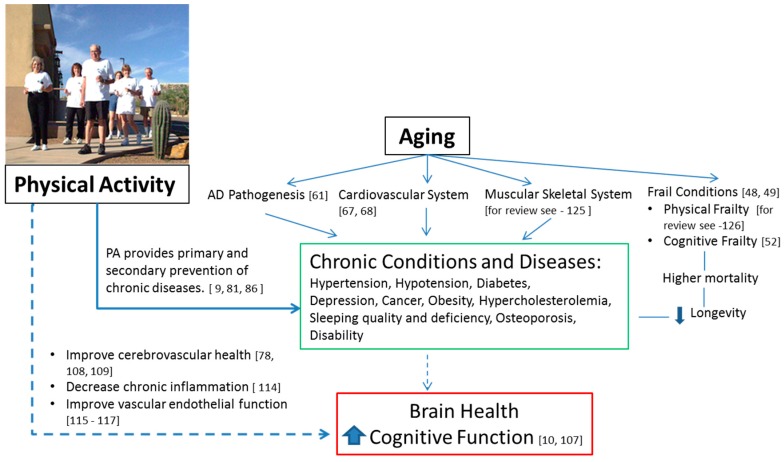
The evidence of PA’s role in reducing the risks of cognitive decline in older adults. The dotted arrows indicate that more research is warranted.

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
