# Peer review of "Physical Activity: A Viable Way to Reduce the Risks of Mild Cognitive Impairment, Alzheimer’s Disease, and Vascular Dementia in Older Adults"

_brainsci, 2017, doi:10.3390/brainsci7020022_

Round 1

Reviewer 1 Report

The Authors reviewed evidence about the associations of physical activity (PA) with major agerelated neurodegenerative diseases and syndromes, including Alzheimer’s disease, vascular dementia, and mild cognitive impairment, also providing evidence of PA’s role in reducing the risks for these diseases for helping to improve cognitive outcomes in older adults. The aim of the review article appears to be clear. However, the Authors should perform some changes or clarification to improve the manuscript:

1. The study has to be completed with some specifications: data pertaining depressive symptoms in the elderly show that they are associated with negative outcomes for physical disabilities. Furthermore, older persons who report depressive symptoms are at higher risk of subsequent physical decline. So the Authors have to specify that the relationship could be complex and bidirectional. (Am J Geriatr Psychiatry. 2009 Feb;17(2):144-54. Physical disability and depressive symptomatology in an elderly population: a complex relationship. The Italian Longitudinal Study on Aging (ILSA).

2. Some finding revealed that genotype such as ApoE alleles modified the association between physical activity and dementia or Alzheimer‘s disease. The apoE epsilon4 carriers may be more vulnerable to environmental factors, and thus, lifestyle interventions may greatly modify dementia risk. The Authors have to better clarify this concept (J Cell Mol Med. 2008 Dec; 12(6B): 2762-71. Epub 2008 Feb 8. Apolipoprotein E epsilon4 magnifies lifestyle risks for dementia: a population-based study).

3. Higher physical activity since midlife was strongly associated with less frailty in old age. Cognitive frailty, a condition describing the simultaneous presence of physical frailty and mild cognitive impairment and the predictive role of a “reversible” cognitive frailty model on incident dementia has been recently. The Authors have to clarify this concept and its importance beyond vascular risk factors and depressive symptoms. (J Am Med Dir Assoc. 2017 Jan;18(1):89.e1-89.e8. doi: 10.1016/j.jamda.2016.10.012. Reversible Cognitive Frailty, Dementia, and All-Cause Mortality. The Italian Longitudinal Study on Aging)

4. Regarding the figures: the quality of the figures have to be improved and the content does not add to the written review.

Author Response

Manuscript#: brainsci-167359

Title: Physical Activity: A Viable Way to Reduce the Risks of Mild Cognitive Impairment, Alzheimer’s Disease, and Vascular Dementia in Older Adults

First, we would like to thank the reviewers for their comments and recommendations. We appreciate the time and effort it takes to review a manuscript and make thoughtful comments and suggestions.  We have responded to these comments. Here we listed our point-by-point responses to the reviewers.  

Overview of Change:

In response to the reviewers’ comments, we have removed the original Figure - Progression of a Neurodegenerative Disorder. We have made the new Figure “to enhance the presentation of this important review” as Reviewer #2 suggested. We have included all the articles that were suggested by reviewers in the manuscript.  

Reviewer #1

The Authors reviewed evidence about the associations of physical activity (PA) with major agerelated neurodegenerative diseases and syndromes, including Alzheimer’s disease, vascular dementia, and mild cognitive impairment, also providing evidence of PA’s role in reducing the risks for these diseases for helping to improve cognitive outcomes in older adults. The aim of the review article appears to be clear. However, the Authors should perform some changes or clarification to improve the manuscript:

1.     The study has to be completed with some specifications: data pertaining depressive symptoms in the elderly show that they are associated with negative outcomes for physical disabilities. Furthermore, older persons who report depressive symptoms are at higher risk of subsequent physical decline. So the Authors have to specify that the relationship could be complex and bidirectional. (Am J Geriatr Psychiatry. 2009 Feb;17(2):144-54. Physical disability and depressive symptomatology in an elderly population: a complex relationship. The Italian Longitudinal Study on Aging (ILSA).

o   This is a great article and gave us a very important point to mention in this article. We had added the ILSA study in the Introduction – Reference # 7.

o   We included the importance of understanding the complex relationship between depression and disability in the section – Preventing/Treating Depression.  

o   In the new Figure we added “Disability” and “Depression” in the Chronic Conditions box, which were mentioned in the ILSA study, “…to understand the complex relationship between depression and disability.”   

2.     Some finding revealed that genotype such as ApoE alleles modified the association between physical activity and dementia or Alzheimer‘s disease. The apoE epsilon4 carriers may be more vulnerable to environmental factors, and thus, lifestyle interventions may greatly modify dementia risk. The Authors have to better clarify this concept (J Cell Mol Med. 2008 Dec; 12(6B): 2762-71. Epub 2008 Feb 8. Apolipoprotein E epsilon4 magnifies lifestyle risks for dementia: a population-based study).

o   We had added the article (Reference #62).

o   In order to explain this study, we first explained APOE ε4 allele and APOE ε4 allele carrier status.  We agree this is a very important concept to understand in dementia research field.  

o   The study by Kivipelto and Colleagues was explained in the Section 3 with other papers (Reference #57- #61). 

3.     Higher physical activity since midlife was strongly associated with less frailty in old age. Cognitive frailty, a condition describing the simultaneous presence of physical frailty and mild cognitive impairment and the predictive role of a “reversible” cognitive frailty model on incident dementia has been recently. The Authors have to clarify this concept and its importance beyond vascular risk factors and depressive symptoms. (J Am Med Dir Assoc. 2017 Jan;18(1):89.e1-89.e8. doi: 10.1016/j.jamda.2016.10.012. Reversible Cognitive Frailty, Dementia, and All-Cause Mortality. The Italian Longitudinal Study on Aging)

o   Another great article by ILSA Working group! Also, it is a great concept – “frailty” to add in our paper!

o   We had added the article (Reference # 53).

o   To readers to understand the concept of cognitive frailty, we explained how it has been proposed and refined. We added a few articles (Reference # 48 - # 52)

o   In the Figure, Physical and Cognitive Frailty were added to indicate as a state of vulnerability.

4.     Regarding the figures: the quality of the figures have to be improved and the content does not add to the written review.

o   We have made a new Figure that shows our main objective of this review paper – to show the evidence of PA’s roles in reducing the risks of cognitive decline in older adults.

Reviewer 2 Report

General Comments:

This is a well written manuscript addressing the important topic of the role of physical activity in reducing the risks of mild cognitive impairment, Alzheimer's disease, and vascular dementia in older adults.  The writing style used in this review is lucid and straightforward. 
The manuscript can be improved by addressing the following:

1) An updated bibliography seems in order, since there have been a number of key studies/systematic reviews published on the same topic in the latter part of 2016 (e.g., Sallis JF, et al. Lancet, 2016;

2) The methods employed in reviewing the literature -- did the authors use a specific systematic reviewing protocol for their review - e.g., rating the strengths of the studies by study design, sampling, effect sizes, quality of exposure and outcome variables, statistical methods, etc.?;

3) Display of findings through the use of tables and/or figures are needed to enhance the presentation of this important review;

Author Response

Manuscript#: brainsci-167359

Title: Physical Activity: A Viable Way to Reduce the Risks of Mild Cognitive Impairment, Alzheimer’s Disease, and Vascular Dementia in Older Adults

First, we would like to thank the reviewers for their comments and recommendations. We appreciate the time and effort it takes to review a manuscript and make thoughtful comments and suggestions.  We have responded to these comments. Here we listed our point-by-point responses to the reviewers.  

Overview of Change:

In response to the reviewers’ comments, we have removed the original Figure - Progression of a Neurodegenerative Disorder. We have made the new Figure “to enhance the presentation of this important review” as Reviewer #2 suggested. We have included all the articles that were suggested by reviewers in the manuscript.  

Reviewer #2:

This is a well written manuscript addressing the important topic of the role of physical activity in reducing the risks of mild cognitive impairment, Alzheimer's disease, and vascular dementia in older adults.  The writing style used in this review is lucid and straightforward. 
The manuscript can be improved by addressing the following:

1.     An updated bibliography seems in order, since there have been a number of key studies/systematic reviews published on the same topic in the latter part of 2016 (e.g., Sallis JF, et al. Lancet, 2016;

o   We had added 20 more papers that are about PA and cognitive decline, including Sallis and his colleagues report article published in Lancet – This is a great report article! (Reference #10).

o   All articles that were recommended by the Reviewer #1 were included.

o   The most recent review article about potential mechanisms of relationship with PA and cognition was added (Journal of Alzheimer’s Disease, 2017, Reference # 107).  

2.     The methods employed in reviewing the literature -- did the authors use a specific systematic reviewing protocol for their review - e.g., rating the strengths of the studies by study design, sampling, effect sizes, quality of exposure and outcome variables, statistical methods, etc.?;

o   We had explained how we reviewed the literature “This review was based on searches of the US National Library of Medicine (PubMed), Ovid MEDLINE, Google Scholar, and Web of Science, using terms to identify the risk exposure (physical inactivity or sedentary) combined with terms to determine the outcomes of interest [cognitive impairment or decline or disorders or AD or dementia or MCI or VaD]. A search filter was developed to include only human studies.”

3.     Display of findings through the use of tables and/or figures are needed to enhance the presentation of this important review.

o   Great point!! We have made a new Figure, titled, “The evidence of PA’s role in reducing the risks of cognitive decline in older adults.”